# Macular Parameters Change during Silicon Oil Tamponade for Retinal Detachment Surgery

**DOI:** 10.3390/medicina59020334

**Published:** 2023-02-10

**Authors:** Sofija Davidović, Maja Živković, Dijana Risimić, Maša Rapajić, Srđan Teodorović, Sava Barišić

**Affiliations:** 1Medical Faculty, University of Novi Sad, 21000 Novi Sad, Serbia; 2Eye Clinic, Clinical Center of Vojvodina, 21000 Novi Sad, Serbia; 3Medical Faculty, University of Niš, 18000 Niš, Serbia; 4Eye Clinic, Clinical Centre Niš, 18000 Niš, Serbia; 5Medical Faculty, University of Belgrade, 11000 Belgrade, Serbia; 6Eye Clinic, Clinical Center of Serbia, 11000 Belgrade, Serbia

**Keywords:** rhegmatogenous retinal detachment, silicon oil tamponade, OCT macular thickness

## Abstract

*Background and Objectives*: To evaluate possible changes in macular thickness parameters during and after silicon oil tamponade and in pars plana rhegmatogenous retinal detachment surgery. *Materials and Methods*: Our retrospective study included 34 consecutive patients who underwent 23-gauge retinal detachment surgery with silicon oil tamponade. Central macular thickness (CMT), central macular volume cube (CMV) and average macular thickness cube (AVG) were measured by optical coherence tomography (OCT) before rhegmatogenous retinal detachment surgery with silicon oil tamponade during tamponade (seven days, one month and three months after surgery), and one month after silicon oil removal. *Results*: In our sample, macular parameters CMT, CMV and AVG in patients who underwent retinal detachment surgery were statistically reduced during silicon oil tamponade (*p* < 0.05). After silicon oil removal, all parameters recovered, reaching numeric values which were not statistically significant compared to preoperative values. The average span of silicon oil tamponade was 162 +/− 23 days. *Conclusions*: Silicon oil tamponade during 23-gauge rhegmatogenous retinal detachment surgery leads to a transitory reduction of central macular thickness, central macular volume cube and average macular thickness cube in eyes that underwent retinal surgery. After silicon oil removal, macular parameters returned to preoperative values in most of the patients.

## 1. Introduction

Retinal detachment (RD) is a relatively urgent and sight-threatening eye condition that can be classified as rhegmatogenous (most frequent), tractional and exudative. The incidence of RD ranges from 6.3 to 17.9 per 100,000 patients [1,2].

Pars plana vitrectomy (PPV) is the preferred surgical treatment for RD. Two tamponading intravitreal agents are commonly used during PPV after vitreous body removal: expanding gas and silicon oil (SO) [3,4]. The choice of a specific tamponading agent depends on the clinical stage and duration of retinal detachment, and SO is routinely used as a tamponading agent for more advanced and complicated cases of RD [5].

Advances in modern ophthalmic surgery and the constant improvement of surgical devices have increased retinal surgery success rate. However, anatomic reattachment of retinal layers during and after the surgery is not the only relevant parameter affecting the overall clinical success of the treatment. Optical coherence tomography (OCT) imaging of delicate anatomic structures of the macula, choroid, retinal nerve fibre layers and the optic nerve head provided new insights into the effects SO can have on these structures during its presence in the eye [6].

In that sense, along with its positive effects of reattaching and keeping the retina in place, silicon oil may manifest specific adverse effects that can influence retinal, choroidal and optic nerve tissues’ health. The underlying pathophysiological mechanisms of ocular tissue damage during SO tamponade are various and still not completely understood.

It has been observed that SO tamponade can lead to transient or permanent retinal ischemia and thinning of inner retinal layers, in the first place of the retinal ganglion cell layer (RGCL), inner plexiform and nuclear layer [7]. Retinal nerve fibre layers of the optic nerve head have also been described as prone to thinning during SO tamponade [8]. Changes have also been recorded in the choroid’s subfoveal region regarding its thickness reduction [9]. During silicon oil tamponade, perfusion is reduced in the macular area, and foveolar avascular zone (FAZ) extension occurs [10]. Therefore, it is essential to emphasise the significance of well-timed silicone oil removal from eyes with surgically reattached retina in order to reduce the rate of possible tamponade-related complications.

Our study aimed to measure and evaluate possible changes in macular thickness parameters during SO tamponade and after its removal in rhegmatogenous retinal detachment surgery.

## 2. Materials and Methods

In this retrospective study, the possible influence of silicon oil on macular thickness was determined in patients who underwent routine rhegmatogenous retinal detachment surgery with silicon oil tamponade. Central macular thickness (CMT), central macular volume cube (CMV) and average macular thickness cube (AVG) were measured by OCT of macula before rhegmatogenous retinal detachment surgery with silicon oil tamponade, during tamponade (seven days, one month and three months after surgery), and one month after silicon oil removal.

Data of 34 patients with rhegmatogenous retinal detachment (30 patients with the macula-off and four patients with macula-on detachment) that had 23-gauge pars plana vitrectomy from January 2021 to January 2022 at the University Eye Clinic of University Clinical Center Vojvodina, in Novi Sad, Serbia, were analysed. All patients received Oxane 1300 Bauch and Lomb silicon oil at the end of 23-gauge three-port pars plana vitrectomy, performed in parabulbar anaesthesia. Only patients with noncomplicated primary retinal detachment forms were included in the study (retinal detachment with proliferative vitreoretinopathy (PVR) grades A and B). This study did not include very fresh RRD treated with SF6 tamponade. Primary RRD that could not be safely attached with SF6 due to a temporary lack of C3F8 supply received silicon oil tamponade. More complex forms of retinal detachment with PVR grades C and D were excluded.

Since cataract formation in the vitrectomised eyes with SO tamponade is common and peripheral vitreous removal is more effective in pseudophakic than in phakic eyes, the combined surgical procedure of cataract removal with foldable acrylic IOL implantation and retinal surgery (phacovitrectomy) was performed in all patients.

Macular parameters change was assessed by OCT device CIRRUS HD-OCT 5000 (ZEISS, Jena, Germany). This study included the following parameters record: central macular thickness in microns (CMT), central macular volume cube in mm^2^ (CMV), and average macular thickness cube in microns (AVG). The operated and fellow eye were included in the follow-up. The parameters were measured preoperatively, seven days after PPV surgery with SO tamponade, 30 days and 90 days after surgery, and 30 days after routine silicon oil removal pars plana surgical procedure in operated and both fellow eyes. Patients with postoperative macular oedema or significant epiretinal membrane were excluded, as were patients with severe retinal or other cellular pathology in the fellow eye.

The axial length of the eye (AXL) and IOL calculation were done preoperatively with a IOL Master 500 non-invasive optical biometer (Carl Zeiss Meditec, Jena, Germany). Best-corrected visual acuity (BCVA) was determined during every visit using standard a Snellen optotype and recorded in decimal notation.

Data acquired were analysed statistically using MedCalc software v 20.104 (MedCalc Software Ltd., Ostend, Belgium) using repeated measures ANOVA with a significance level of *p* < 0.05. Results were presented as tables.

## 3. Results

This study involved 34 patients who underwent retinal detachment surgery with silicon oil tamponade. There were 30 patients with the macula-off and four patients with macula-on retinal detachment. A single vitreoretinal surgeon did all phacovitrectomy surgeries. The average span of silicon oil tamponade was 162 +/− 23 days.

In our sample, the average age of patients with rhegmatogenous retinal detachment that were operated on was 45 years, ranging from 23 to 72 years. The gender proportion was 41% female and 59% male patients.

Preoperatively, the mean best-corrected visual acuity was 0.22, with the lowest value seven days after the surgery. After day seven, BCVA continually improved. There was a statistically significant increase in BCVA between preoperative and values after SO removal (Table 1).

The arithmetic mean of AXL in our patients was 24.15 mm (SD = 2.40/95% CI = 23.31 to 24.99), with a maximum value of 31.49 mm and a minimum value of 20.60 mm.

### 3.1. Central Macular Thickness—CMT (μm)

Table 2 and Table 3 present the CMT values recorded preoperatively, seven days, 30 days and 90 days after ophthalmic surgery with SO tamponade, as well as 30 days after silicon oil removal (SOR), both in operated and non-operated fellow eyes.

Our sample data analysis showed that CMT was significantly reduced at days 7, 30 and 90 after PPV SO surgery, compared to preoperatively measured values.

Thirty days after SOR, CMT values recovered to initial preoperative values, with no statistical significance between starting preoperative values and those 30 days after SOR (Table 2).

Table 3 shows that in the non-operated, fellow eye no significant change of CMT was noted in the consecutive period.

### 3.2. Central Macular Volume Cube—CMV (mm^3^)

Table 4 and Table 5 present CMV data recorded preoperatively, seven days, 30 days and 90 days after ophthalmic surgery with SO tamponade, and 30 days after silicon oil removal (SOR), both in operated and in non-operated fellow eyes.

Our data analysis showed that CMV was significantly reduced at days 30 and 90 after PPV SO surgery, compared to preoperatively recorded values.

Thirty days after SOR, measurements of CMV were almost equal to initial preoperative values, with no statistical significance between starting preoperative values and those 30 days after SOR (Table 4).

Table 5 shows that in the non-operated, fellow eye a significant change of CMV was also present 7 and 90 days after the surgery. However, no statistically significant difference in the central macular volume cube in the fellow eye was found before surgery and 30 days after SO removal.

### 3.3. Average Macular Thickness Cube—AVG (µm)

Table 6 and Table 7 summarise the average macular thickness cube (AVG) data. The data were collected preoperatively, seven days, 30 days and 90 days after ophthalmic surgery with SO tamponade, and 30 days after silicon oil removal (SOR), both in operated and non-operated fellow eyes.

In our analysed patients’ sample, AVG was significantly reduced at days 7, 30 and 90 after PPV SO surgery, compared to preoperatively measured values.

Thirty days after SOR, measurements of AVG had almost recovered to initial preoperative values, but with statistical significance between starting preoperative values and those 30 days after SOR (Table 5).

Table 7 shows that in the non-operated, fellow eye no significant change of AVG was noted in the presented periods of consecutive average macular thickness cube analysis, except seven days after retinal detachment surgery with silicon oil tamponade.

The parameters recorded by OCT were central macular thickness in microns (CMT), central macular volume cube in mm^2^ (CMV), and average macular thickness cube in microns (AVG) in patients with PPV retinal detachment surgery and silicone oil tamponade. Here, we present OCT scans of a patient with macula-on type of retinal detachment on their right eye preoperatively, one month after PPV with SO tamponade and postoperatively after SO removal (Figure 1).

## 4. Discussion

Despite the introduction of novel therapeutic strategies in treating rhegmatogenous retinal detachment patients, including expandible gases as tamponading agents during pars plana vitrectomy surgery, silicon oil still has an important place, especially in more clinically difficult RD cases. However, SO could have many severe and unwanted effects on all parts of the eye [11]. Advanced diagnostic possibilities, such as the implementation of OCT of the macula and retinal nerve fibre layer in the everyday practice of posterior segment clinics, give us essential information that can be very useful for both vitreoretinal surgeons and ophthalmic scientists.

Silicon oil-induced delicate retinal changes can explain possible visual acuity loss in operated patients with no underlying clinical entity [12]. The central macular thickness (CMT) may reduce during SO tamponade and recover after SO evacuation in operated eyes with RD. Changes in retinal layer thickness are described in many studies. Macular thinning is explained by the reduction of inner retinal layers, in particular, leading to subsequent lowering or central vision loss [13,14].

Rabina G. et al. reported a decrease in macular layer thickness and visual acuity in their study of 41 patients with PPV SO surgery for RD [15]. The average age of patients in their sample was 56 years, while in our sample the mean age of operated patients with rhegmatogenous RD was 45 years. The average time of SO tamponade duration in the paper of Rabina et al. was 151 days, compared to our data which included patients with an average duration of 162 days. The central macular thickness of the operated eye in the paper of Rabina G. et al. increased from 249 ± 50 µm before to 279 ± 48 µm after SO removal (*p* < 0.001), compared to 281 ± 21 µm in the fellow eye (*p* < 0.001). Our patients demonstrated an increase in CMT on the 90th day after PPV SO surgery from 244.76 microns to 261.79 average value 30 days after SO evacuation (*p* < 0.05). A possible explanation of these changes may be the recovery of SO-induced damage to the retinal microstructure after SOR.

Hong et al. reported changes in retinal vessels and layer thickness in RD surgery. They included 20 eyes with macula-off RD and 11 eyes with macula-on RD. In the macula-off RD group, the central retinal thickness was significantly decreased six months post-operation compared with the fellow eyes (228.9 ± 29.7 µm and 253.6 ± 27.7 µm, *p* = 0.009) [16]. In this paper, we did not analyse data separately for macula-off and macula-on groups of RD patients, which could be a valuable addition to future studies.

Zhou and colleagues analysed the effects of different tamponading agents (SO and gas) on retinal thickness change and macular perfusion after PPV RD surgery. Their study included 21 operated eyes, and compared to gas tamponade, SO led to a more significant decrease in both superficial and deep retinal blood flow, confirming once more that compared to gas, silicon oil could have more negative tamponade effects on fundus vasculature [17]. Similar results with macular vessel density reduction and central retinal thickness reduction during SO tamponade were found in the paper of Liu Y et al. [18].

Roohipoor et al. contributed to the topic of SO tamponade and its possible adverse effects on fine retinal and choroidal structures with their paper, including the measurement of macular OCT angiography (OCTA-A) in 45 operated eyes [19]. Their findings suggested that vascular density of the parafoveal plexus demonstrated a statistically significant decrease in the postoperative silicon-filled eye, compared to the fellow eye, along with a decrease in central macular thickness. After SO removal, some improvement in these measurements was recorded, but they remained reduced compared to the fellow eye.

The presented results of our study included 34 eyes of patients with rhegmatogenous retinal detachment who were treated with 23-gauge PPV surgery and received silicon oil tamponade to reattach the retina in a routine surgical procedure. It has been observed that SO tamponade can lead to a thinning of the transient retinal layer and a decrease in central macular thickness and volume. However, the underlying mechanisms are still not known. Mechanical effects of SO with its surface tension, transient change of retinal metabolism, the toxic effect of SO, or changes in microperfusion of the retina, subfoveal choroid, and retinal nerve fibre layers with subsequent hyponutrition and ischaemia may be contributing factors.

Restoring most of the macular parameters to the values approximating those before the surgery, which were recorded in this study, emphasises once more the significance of timely silicone oil removal (SOR) from eyes with attached retina to reduce the rate of possible complications.

Retinal detachment surgery is critical in everyday ophthalmo-surgical practice. Thanks to the quick evolution of technology and devices applied, new approaches to data interpretation and unique solutions for operating procedures and tamponading agents are expected.

## 5. Conclusions

Silicon oil tamponade during routine 23-gauge rhegmatogenous retinal detachment surgery leads to changes of central macular parameters in eyes that have undergone retinal surgery. Transitory reduction of central macular thickness, average macular thickness cube and change of central macular thickness cube are recorded, but in the literature available the precise mechanism is still unknown. However, most macular parameters return to a similar preoperative value after silicon oil removal.

In this regard, further and more complex data analysis is required, including measurement and interpretation of macular thickness changes in correlation with choroidal subfoveal thickness change, peripapillary retinal nerve fibre change and choroidal capillary perfusion change during silicon oil tamponade in retinal detachment surgery.

## Figures and Tables

**Figure 1 medicina-59-00334-f001:**
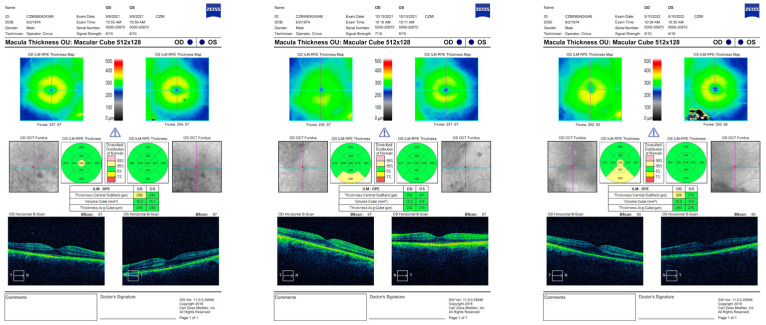
OCT scan printouts of a patient with macula-on type of RD on right eye preoperatively, one month postoperatively and six months after SO removal (from left to right).

**Table 1 medicina-59-00334-t001:** Best-corrected visual acuity (BCVA) in operated eyes (Snellen).

		n	Variable 1		Variable 2		Paired Differences			
Pair			Mean	SD	Mean	SD	Mean	SD	95% CI	*p* ^a^
BCVA preop	BCVA 7 days	34	0.22	0.33	0.18	0.19	0.04	0.04	−0.07 to 0.17	1.0000
BCVA preop	BCVA 1 month	34	0.22	0.33	0.22	0.18	0.005	0.04	−0.14 to 0.15	1.0000
BCVA preop	BCVA 3 months	34	0.22	0.33	0.29	0.23	−0.06	0.05	−0.22 to 0.08	1.0000
BCVA preop	BCVA post SO ex	34	0.22	0.33	0.41	0.28	−0.18	0.05	−0.35 to −0.01	0.0311

^a^ Repeated measures ANOVA.

**Table 2 medicina-59-00334-t002:** Central macular thickness—CMT (µm) in PPV SO operated eyes.

		n	Variable 1		Variable 2		Paired Differences			
Pair			Mean	SD	Mean	SD	Mean	SD	95% CI	*p* ^a^
CMT OP preop	CMT OP 7 days	34	266.47	50.84	238.38	54.63	28.088	5.127	12.666 to 43.511	<0.0001
CMT OP preop	CMT OP 1 month	34	266.47	50.84	242.71	53.29	23.765	4.522	10.160 to 37.369	0.0001
CMT OP preop	CMT OP 3 months	34	266.47	50.84	244.76	51.65	21.706	4.360	8.589 to 34.823	0.0002
CMT OP preop	CMT OP post SO ex	34	266.47	50.84	261.79	50.05	4.676	6.290	−14.245 to 23.598	1.0000

^a^ Repeated measures ANOVA.

**Table 3 medicina-59-00334-t003:** Central macular thickness—CMT (µm) in non-operated eyes.

		n	Variable 1		Variable 2		Paired Differences			
Pair			Mean	SD	Mean	SD	Mean	SD	95% CI	*p* ^a^
CMT Non-OP preop	CMT Non-OP 7 days	34	268.55	36.17	266.50	35.89	2.05	1.95	−3.81 to 7.93	1.0000
CMT Non-OP preop	CMT Non-OP 1 month	34	268.55	36.17	267.61	36.44	0.94	2.24	−5.80 to 7.68	1.0000
CMT Non-OP preop	CMT Non-OP 3 months	34	268.55	36.17	271.79	38.96	−3.23	2.67	−11.28 to 4.81	1.0000
CMT Non-OP preop	CMT Non-OP post SO ex	34	268.55	36.17	271.26	38.10	−2.70	2.68	−10.78 to 5.37	1.0000

^a^ Repeated measures ANOVA.

**Table 4 medicina-59-00334-t004:** Central macular volume cube CMV (mm3) in PPV SO operated eyes.

		n	Variable 1		Variable 2		Paired Differences			
Pair			Mean	SD	Mean	SD	Mean	SD	95% CI	*p* ^a^
CMV OP preop	CMV OP 7 days	33	10.41	1.78	9.90	1.41	0.51	0.19	−0.08 to 1.10	0.1442
CMV OP preop	CMV OP 1 month	33	10.41	1.78	9.64	1.43	0.77	0.24	0.02 to 1.51	0.0374
CMV OP preop	CMV OP 3 months	33	10.41	1.78	9.51	1.41	0.90	0.23	0.20 to 1.59	0.0045
CMV OP preop	CMV OP post SO ex	33	10.41	1.78	10.03	1.80	0.38	0.28	−0.46 to 1.22	1.0000

^a^ Repeated measures ANOVA.

**Table 5 medicina-59-00334-t005:** Central macular volume cube—CMV (mm3) in non-operated eyes.

		n	Variable 1		Variable 2		Paired Differences			
Pair			Mean	SD	Mean	SD	Mean	SD	95% CI	*p* ^a^
CMV Non-OP preOP	CMV Non-OP 7 days	34	10.05	0.82	9.76	1.11	0.29	0.08	0.02 to 0.56	0.0256
CMV Non-OP preOP	CMV Non-OP 1 month	34	10.05	0.82	9.87	0.97	0.18	0.08	−0.06 to 0.43	0.3347
CMV Non-OP preOP	CMV Non-OP 3 months	34	10.05	0.82	9.75	1.10	0.30	0.08	0.03 to 0.57	0.0185
CMV Non-OP preOP	CMV Non-OP post SO ex	34	10.05	0.82	9.89	1.07	0.16	0.08	−0.08 to 0.40	0.5256

^a^ Paired samples *t*-test.

**Table 6 medicina-59-00334-t006:** Average macular thickness cube—AVG (µm) in PPV SO operated eyes.

		n	Variable 1		Variable 2		Paired Differences			
Pair			Mean	SD	Mean	SD	Mean	SD	95% CI	*p* ^a^
AVG OP preop	AVG OP 7 days	34	285.38	45.58	262.94	43.66	22.44	4.15	9.93 to 34.94	0.0001
AVG OP preop	AVG OP 1 month	34	285.38	45.58	259.61	45.69	25.76	3.60	14.91 to 36.61	<0.0001
AVG OP preop	AVG OP 3 months	34	285.38	45.58	260.50	47.04	24.88	4.10	12.53 to 37.22	<0.0001
AVG OP preop	AVG OP post SO ex	34	285.38	45.58	276.35	42.87	9.02	2.81	0.56 to 17.49	0.0296

^a^ Repeated measures ANOVA.

**Table 7 medicina-59-00334-t007:** Average macular thickness cube—AVG (µm) in non-operated eyes.

		n	Variable 1		Variable 2		Paired Differences			
Pair			Mean	SD	Mean	SD	Mean	SD	95% CI	*p* ^a^
AVG Non-OP preop	AVG Non-OP 7 days	34	274.44	29.14	273.05	28.10	1.38	0.47	−0.03 to 2.80	0.0613
AVG Non-OP preop	AVG Non-OP 1 month	34	274.44	29.14	273.11	28.72	1.32	0.95	−1.55 to 4.19	1.0000
AVG Non-OP preop	AVG Non-OP 3 months	34	274.44	29.14	274.47	30.49	−0.02	1.24	−3.75 to 3.70	1.0000
AVG Non-OP preop	AVG Non OP post SO ex	34	274.44	29.14	274.38	30.52	0.05	1.15	−3.40 to 3.52	1.0000

^a^ Paired samples *t*-test.

## Data Availability

Data supporting results presented in the study can be enclosed if needed or requested by the reviewer.

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
