# Peer review of "Macular Parameters Change during Silicon Oil Tamponade for Retinal Detachment Surgery"

_medicina, 2023, doi:10.3390/medicina59020334_

Round 1

Reviewer 1 Report

Congratulations for your work! Indeed, retinal detachment is a condition that affects greatly one’s life and is responsible for loss of quality of vision. Despite several developments made in the last decades, there’s still a long way to go. Your work is very interesting as it addresses the potential effects of the treatment on retinal structure. I have some questions and suggestions:

Introduction

 - line 63 to 67 – this is better placed in Methods

Methods:

 - were those primary RRD? If so and as all were uncomplicated, why did you choose silicone oil as primary intention tamponade?

 - how could you accurately measure the retinal parameters before the surgery? All maculas were attached?

 - line 80 and 81 – this information belongs to the Results section

 - line 86 – Excel

 - line 92 to 94 – this information belongs to the Results section; in addition, you’re presenting gender proportion, not ratio

 - the mean age of your population was 45 years, ranging from 23 to 72 years. Why did you perform cataract surgery in all patients?

 - why did you choose a paired t-test comparison instead of a repeated measures ANOVA?

Results

 - where is your Table 1 describing the population? You’ve stated the age and gender but that is way too minimalist. Please consider presenting the preoperative visual acuity, spherical equivalent or axial length, macular status, among others

 - line 166 to 178 – this belongs to the Discussion

Discussion

 - line 190 – exactly! Where’s your visual acuity data?

Author Response

Introduction

 - line 63 to 67 – this is better placed in Methods

Corrected.

Methods:

 - were those primary RRD? If so and as all were uncomplicated, why did you choose silicone oil as primary intention tamponade? 

There were all primary RRD included in this study sample. Very fresh RRD that were treated with SF6 tamponade, were not included in this study. At certain period of time tamponade C3F8 was not available at our clinic, due to problems of supply (problems with suppliers…). Therefore, primary RRD that seemed that could be not safely attached with SF6, received silicon oil tamponade.

 - how could you accurately measure the retinal parameters before the surgery? All maculas were attached?

There were more retinal detachment surgeries going on, but only patients where we were able to measure and follow up macular parameters (both before and after surgery) were included in the study.

 - line 80 and 81 – this information belongs to the Results section

Corrected.

 - line 86 – Excel

Corrected.

 - line 92 to 94 – this information belongs to the Results section; in addition, you’re presenting gender proportion, not ratio

Corrected.

 - the mean age of your population was 45 years, ranging from 23 to 72 years. Why did you perform cataract surgery in all patients?

It is known that there is higher incidence of cataract formation in eyes with silicon oil tamponade. Also, meticulous peripheral vitreous removal is easier and more achievable with intraocular lens, then with natural lens (easy to get lens touch and more rapid opacification).

 - why did you choose a paired t-test comparison instead of a repeated measures ANOVA?

We have accepted your suggestion and  decided to use repeated measures ANOVA.

Results

 - where is your Table 1 describing the population? You’ve stated the age and gender but that is way too minimalist. Please consider presenting the preoperative visual acuity, spherical equivalent or axial length, macular status, among others

We added detailed demographic data in the Results section.

 - line 166 to 178 – this belongs to the Discussion

Corrected.

Discussion

 - line 190 – exactly! Where’s your visual acuity data?

We added the BCVA to our results section.

Reviewer 2 Report

The authors present a study on the changes in macular characteristics on OCT before and after silicone oil injection. 

I have a few concerns:

1. As pointed out by the authors, there already have been several studies on this topic. Thus it needs to be clear on what extra information does this study provide.

2. How did the authors calculate the macular thickness and volume in a detached retina (preop parameters)? Due to the oblique orientation of the detached retina, a significant error can occur during measurement (especially volume measurements). The elongated photoreceptor outer segments and outer retinal folds in the detached retinal can also overestimate the thickness. Kindly clarify on these points.

3. It would be good to illustrate how the measurements were done with the help of a figure.

4. Did the macular parameters have any influence on the post-operative visual recovery?

Author Response

  1. As pointed out by the authors, there already have been several studies on this topic. Thus it needs to be clear on what extra information does this study provide.

Rhegmatogenous retinal detachment surgery is frequent, and despite many advances in surgical practice, technology and machines that we use, new tamponading agents, etc., it still has some sides that are still completely unknown to us. Studies about the effect of silicon oil on retina are available, but still are certain scientific doubts and data that are not in complete accordance… We wanted to make our own study with personal experience, and contribute to overall knowledge available. And as well to make entry for our further analyses and studies that could follow.

  1. How did the authors calculate the macular thickness and volume in a detached retina (preop parameters)? Due to the oblique orientation of the detached retina, a significant error can occur during measurement (especially volume measurements). The elongated photoreceptor outer segments and outer retinal folds in the detached retinal can also overestimate the thickness. Kindly clarify on these points.

There were other primary RRD surgeries performed during the data collection for this study. In order to try to obtain correct samples of OCT of macula in patients with RRD, we selected only patients where macula was possible to record both preoperatively and after surgery, to be able to make reasonable comparison. Results with completely detached macula or retina, and extreme values were not included in the study sample.

  1. It would be good to illustrate how the measurements were done with the help of a figure.

Please clarify what kind of illustration you ment us to have.

  1. Did the macular parameters have any influence on the post-operative visual recovery?

OCT measurement of preoperative and postoperative macular area can suggest the possible outcome of RRD surgery in termes of postoperative visual acuity. Limitations of this study are relatively small sample, and short period of postoperative follow up.  Further investigations are needed to determine the proper relationship of OCT recordings and postoperative visual acuity recovery, possibly including OCT A as well. Special challenge presents RRD with detached (bullous) retina.

Round 2

Reviewer 2 Report

Queries have been well addressed.

Kindly provide an image demonstrating preoperative and post-operative retinal thickness measurements.

Author Response

Thank you for your suggestions.

We inserted the image presenting preoperative and postoperative retinal thickness measurements.